# Paraphrase Types for Generation and Detection

**Jan Philip Wahle, Bela Gipp, Terry Ruas**
University of Göttingen, Germany
{wahle,gipp,ruas}@uni-goettingen.de

## Abstract

Current approaches in paraphrase generation and detection heavily rely on a single general similarity score, ignoring the intricate linguistic properties of language. This paper introduces two new tasks to address this shortcoming by considering *paraphrase types* - specific linguistic perturbations at particular text positions. We name these tasks Paraphrase Type Generation and Paraphrase Type Detection. Our results suggest that while current techniques perform well in a binary classification scenario, i.e., paraphrased or not, the inclusion of fine-grained paraphrase types poses a significant challenge. While most approaches are good at generating and detecting general semantic similar content, they fail to understand the intrinsic linguistic variables they manipulate. Models trained in generating and identifying paraphrase types also show improvements in tasks without them. In addition, scaling these models further improves their ability to understand paraphrase types. We believe paraphrase types can unlock a new paradigm for developing paraphrase models and solving tasks in the future.

## 1 Introduction

Paraphrases are texts expressing identical meanings that use different words or structures (Vila et al., 2015, 2014; Zhou and Bhat, 2021). Paraphrases exhibit humans' complex language's nature and diversity, as there are infinite ways to transform one text into another without altering its meaning. For example, one can change a text's

> *morphology*: "Who they **could** ~~might~~ be?",
>
> *syntax*: "~~He drew~~ a go **was drawn** by him.",
>
> *lexicon*: "She ~~liked~~ **enjoyed** it.".

Nonetheless, current paraphrase generation and detection systems are yet unaware of the lexical variables they manipulate (Zhou and Bhat, 2021).

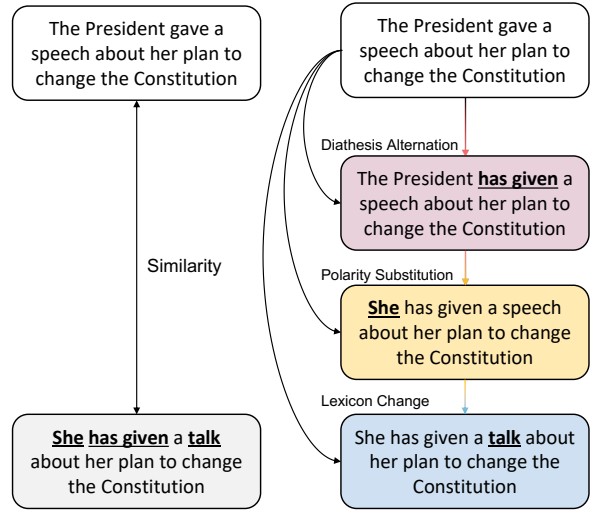

Figure 1: Comparison of current paraphrase tasks (left) and our proposal towards paraphrase types (right).

Generative models cannot be asked to perform certain types of perturbations, and detection models are unable to understand which paraphrase types they detect (Egonmwan and Chali, 2019; Meng et al., 2021; Ormazabal et al., 2022; Vizcarra and Ochoa-Luna, 2020), or they learn limited language aspects (e.g., primarily syntax (Chen et al., 2019; Goyal and Durrett, 2020; Huang and Chang, 2021)). The shallow notion of what composes paraphrases used by these systems limits their understanding of the task and makes it challenging to interpret detection decisions in practice. For example, although high structural and grammatical similarities can indicate plagiarism, detection systems are often not concerned with the aspects that make two texts segments alike (Ostendorff et al., 2022; Wahle et al., 2022a).

Given the limitations of current paraphrase generation and detection tasks, their proposed solutions are also constrained (Foltýnek et al., 2020). Figure 1 gives an example of the difference between current tasks and their true linguistic diversity. Therefore, we propose two new tasks

to explore the role of paraphrase types, namely **Paraphrase Type Generation** and **Paraphrase Type Detection**. In the generation task, a model has to generate a paraphrased text for specific segments considering multiple paraphrase types (§3.1). For the detection task, paraphrased segments must be classified into one or more paraphrase types (e.g., lexico-syntactic-based changes) (§3.2). These tasks complement existing ones (without paraphrase types), enabling more granular assessments of paraphrased content.

The shift from traditional paraphrase tasks towards a more specific scenario, i.e., including paraphrase types, has many benefits. A direct consequence of incorporating paraphrase types, and maybe the most impactful, lies in plagiarism detection. Plagiarism detection systems based on machine learning often support their decision results using shallow high-level similarity metrics that only indicate how much a given text is potentially plagiarized, thus limiting their analysis (Foltýnek et al., 2019). Incorporating paraphrase types allows for more interpretable plagiarism detection systems, as more informative and precise results can be derived. Additionally, automated writing assistants can be improved beyond simple probability distributions when suggesting alterations to a text. On top of that, second-language learners can correct their texts by considering specific paraphrase types in their daily lives (e.g., learning about contractions "does not = doesn't"), helping them to learn new languages faster.

Our proposed tasks indicate that language models struggle to generate or detect paraphrase types with acceptable performance, underlining the challenging aspect of finding the linguistic aspects within paraphrases. However, learning paraphrase types is beneficial in generation and detection as the performance of trained models consistently increases for both tasks. In addition, scaling models also suggest improvements in their ability to understand and differentiate paraphrase types when transferring to unseen paraphrasing tasks.

In summary, we:

- introduce two new tasks, Paraphrase Type Generation and Paraphrase Type Detection, providing a more granular perspective over general similarity-based tasks for paraphrases;

- show that our proposed tasks are compatible with traditional paraphrase generation and detection tasks (without paraphrase types);

- investigate the correlation between paraphrase types, generation and detection performance of existing solutions, and scaling experiments to explore our proposed tasks;

- make the source code and data to reproduce our experiments publicly available;[1]

- provide an interactive demo to generate paraphrases with types;[2]

## 2 Related Work

First attempts to categorize the lexical variables manipulated in paraphrases into a taxonomy have been performed by Vila et al. (2014), followed by their first typology annotated corpus (Vila et al., 2015). Gold et al. (2019) categorize paraphrases on a higher level as meaning relations and present three additional categories: textual entailment, specificity, and semantic similarity. Kovatchev et al. (2020) extend Vila et al. (2015, 2014)'s typology and re-annotate the MRPC-A (Dolan and Brockett, 2005) corpus with fine-grained annotations with more than 26 lexical categories, such as negation switching and spelling changes in the ETPC dataset. Recent works model objective functions instead of taxonomies (e.g., word position deviation, lexical deviation) to automatically categorize paraphrases (Liu and Soh, 2022a). This approach is similar to Bandel et al. (2022); Liu et al. (2020a)'s proposed metrics for paraphrase quality (e.g., semantic similarity, expression diversity).

Recent work requires texts to satisfy certain stylistic, semantic, or structural requirements, such as using formal language or expressing thoughts using a particular template (Iyyer et al., 2018; Shen et al., 2017). In paraphrase generation, methods require texts to meet certain quality criteria, such as semantic preservation and lexical diversity (Bandel et al., 2022; Yang et al., 2022) or require syntactic criteria, such as word ordering (Chen et al., 2019; Goyal and Durrett, 2020; Sun et al., 2021). The development of the Multi-Topic Paraphrase in Twitter (MultiPIT) corpus addresses quality issues in existing paraphrase datasets and facilitates the acquisition and generation of high-quality paraphrases (Dou et al., 2022). Parse-Instructed Prefix (PIP) tunes large pre-trained language models for

---

[1] https://github.com/jpwahle/emnlp23-paraphrase-types

[2] https://huggingface.co/spaces/jpwahle/paraphrase-type-generation

generating paraphrases according to specified syntactic structures in a low-data setting, significantly reducing training costs compared to traditional fine-tuning methods (Wan et al., 2023).

Although current contributions to generate and detect different paraphrase forms, they do not use them to generate or detect paraphrase types directly. Instead, they rely on shallow similarity measures and binary labels for identifying paraphrases. In this work, we propose two new tasks. One for generating specific perturbations when creating new paraphrases and one for detecting the lexical differences between paraphrases. We use the ETPC dataset to evaluate these tasks and show that learning paraphrase types is more challenging than considering the binary notion of paraphrase. Our results suggest that learning paraphrase types is beneficial for traditional paraphrase generation and detection. We further demonstrate these findings in our experiments (§4).

## 3 Task Formulation

Most paraphrase-related tasks focus on generating or classifying paraphrases at a general level (Foltýnek et al., 2020; Wahle et al., 2022a,b, 2021). This goal is limited, as it provides little details on what composes a paraphrase or which aspects make original and paraphrase alike (Fournier and Dunbar, 2021). We believe incorporating paraphrase types in generation- and detection-related tasks can help understand paraphrasing better.

We propose specific tasks for paraphrase generation and detection to include paraphrase types. The goal of **Paraphrase Type Generation** is to generate a paraphrased text that preserves the semantics of the source text but differs in certain linguistic aspects. These linguistic aspects are specific paraphrase types (e.g., lexicon change). In the **Paraphrase Type Detection** task, the goal is to locate and identify the paraphrase types in which two pieces of the text differ.

Both tasks aim to include a fine-grained understanding of paraphrase types over current existing tasks. Each specific task can also be formulated as a simple paraphrase generation or detection task (i.e., without paraphrase types) with a small error $\epsilon$. Thus, our tasks complement existing ones in paraphrase generation and detection. Section 4 provides more information on the composition of the proposed datasets, their splits, and their structure. Figure 2 illustrate our proposed tasks.

### 3.1 Paraphrase Type Generation

Given a sentence or phrase $x$ and a set of paraphrase type(s) $l_i \in L$, a paraphrase $\tilde{x}$ should be provided, where $L$ is the set of all possible paraphrase types $L = \{l_{lex}, ..., l_{morph}\}$. The reference paraphrase types $l_i$ to be incorporated in $\tilde{x}$ have to take place on specific positions (i.e., segments $s_i$), which can potentially overlap. The resulting paraphrase $\tilde{x}$ should maximize its similarity against the original text $x$ while incorporating the segment's reference paraphrase type(s).

The task can be measured through multiple metrics. This study uses BLEU (Papineni et al., 2002), ROUGE (Lin, 2004), and BERTScore (Zhang et al., 2020b) for the paraphrase segments in $\tilde{x}$ in relation to $x$ (cf. Section 4.1). To measure correlations, we also include word position deviation and lexical deviation (Liu and Soh, 2022a),

### 3.2 Paraphrase Type Detection

Given a sentence or phrase $x$ and a paraphrase $\tilde{x}$, the task is to identify which paraphrase types $l_i \in L$ the latter contains in relation to the former $L(\tilde{x})$, where $L$ is the group of all possible paraphrase types $L = \{l_{lex}, ..., l_{morph}\}$ and $L(\tilde{x})$ represents the paraphrase types in $\tilde{x}$. Both $x$ and $\tilde{x}$ include no information about which segments $s_j$ were altered or how they are correlated. Therefore, our task requires the identification of segments and their classification, which can be composed of multiple types. $s_j$ might have different positions in $x$ and $\tilde{x}$, as each phrase can have different word order. The following example shows how two phrases are related according to their paraphrase types.

$x$: A project $_{s_1}$ was funded $_{s_2}$ in $_{s_3}$ **New York City** $_{s_4}$.

$\tilde{x}$: **New York** $_{s_4}$ funded $_{s_2}$ it $_{s_1}$ for $_{s_3}$ **its largest city** $_{s_4}$.

$L(\tilde{x})$: $\{(s_1, l_{lex}); (s_2, l_{syn}); (s_3, l_{dis}); (s_4, l_{lex})\}$

Multiple metrics can also be considered when evaluating Paraphrase Type Detection (e.g., F1 score, accuracy). We evaluate the detection as a weighted sum of accuracies of paraphrase types $l_i$ in the modified segments of $\tilde{x}$ against $x$. Therefore, accuracies are weighted within the same phrase. Weighting prevents the dominance of specific types in phrases with multiple occurrences in the dataset.

As our goal is to explore paraphrase types, we assume that one of the sentences is a paraphrase of the other and both are semantically related. However, this task can also be altered to identify the existence

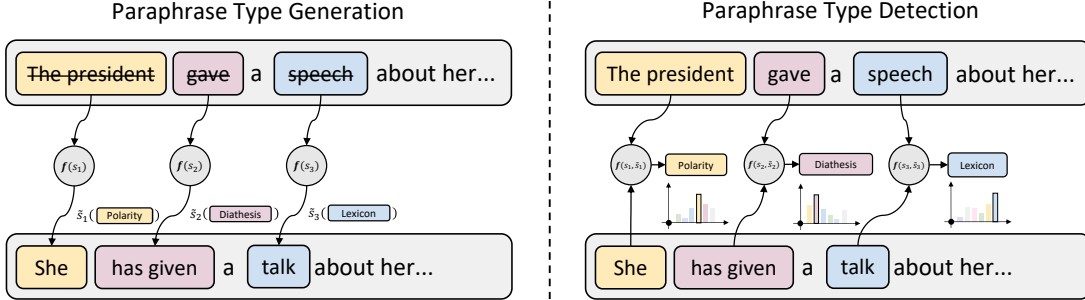

Figure 2: Paraphrase Type Generation (left) and Paraphrase Type Detection (right) with model $f$, reference segments $s1, s2, s3$ and candidate segments $\tilde{s}_1, \tilde{s}_2, \tilde{s}_3$.

of paraphrasing and its types separately. Another possible extension for our tasks could include unrelated sentence pairs with no paraphrase types, similar to the unanswerable questions in SQuAD 2.0 (Li, 2019). In our experiments, we do not investigate the performance considering the location of each paraphrase. Thus, a correct identification is only considered if the pair $(s_j, l_i)$ is provided. We leave the investigation of such aspects to future work and invite researchers to explore other variations of our tasks.

## 4 Experiments

In our experiments, we first investigate the distinct differences in paraphrase types by evaluating their correlations. Next, we measure how much language models already know about paraphrase types and how that changes when scaling them. Finally, we empirically study how existing models perform in our proposed tasks. To test whether paraphrase types are a valuable extension to traditional paraphrasing tasks, we evaluate selected models after incorporating paraphrase types in their training to quantify the transfer from paraphrase types to traditional paraphrase tasks.

### 4.1 Setup

**Datasets.** We use the Extended Paraphrase Typology Corpus (ETPC) (Kovatchev et al., 2018) and three challenging auxiliary paraphrase datasets according to (Becker et al., 2023): Quora Question Pairs (QQP) (Wang et al., 2017), Twitter News URL Corpus (TURL) (Lan et al., 2017), and Paraphrase Adversaries from Word Scrambling (PAWS) (Zhang et al., 2019). More details about the datasets can be found in Appendix A.3. ETPC is a corpus with binary labels (paraphrased or original) and 26 fine-grained paraphrase types, and six high-level paraphrase groups. Table 1 gives an

overview of their distribution. The most common is the group "lexicon-based changes", particularly the paraphrase type "synthetic/analytic substitutions", e.g., noun replacements with the same meaning. Notably, many paraphrases are additions or deletions of words to a phrase. Using ETPC, we evaluate how well existing models perform generation and detection tasks, with and without prior training in paraphrase types. We use a 70% train and 30% eval split with an equal balance between paraphrase types. To show how our tasks are compatible with general binary paraphrase tasks and datasets, selected models trained with paraphrase types are tested for paraphrase types (ETPC) and a general paraphrase task (QQP).

**Metrics.** For the Paraphrase Type Generation task, we measure the performance of models generating paraphrase types on a segment level using BLEU and ROUGE. For Paraphrase Type Detection, we use accuracy on a segment level, i.e., each segment receives an individual score which we average per phrase for three categories: *Binary* - paraphrased or not; *Type* - paraphrase type (e.g., ellipsis); and *Group* - groups of paraphrase types (e.g., morphology-based changes). More details on the evaluation can be found in the Appendix A.5.

**Models.** We conduct experiments scaling model sizes with LLaMA (Touvron et al., 2023); generation experiments with autoregressive models: BART (Lewis et al., 2020), PEGASUS (Zhang et al., 2020a) and ChatGPT; and detection experiments with autoencoders: BERT (Devlin et al., 2019), RoBERTa (Liu et al., 2019), ERNIE 2.0 (Sun et al., 2020), DeBERTa (He et al., 2021), and ChatGPT. We use ChatGPT in the September 25th 2023 version[3].

---

[3] https://help.openai.com/en/articles/6825453-chatgpt-release-notes

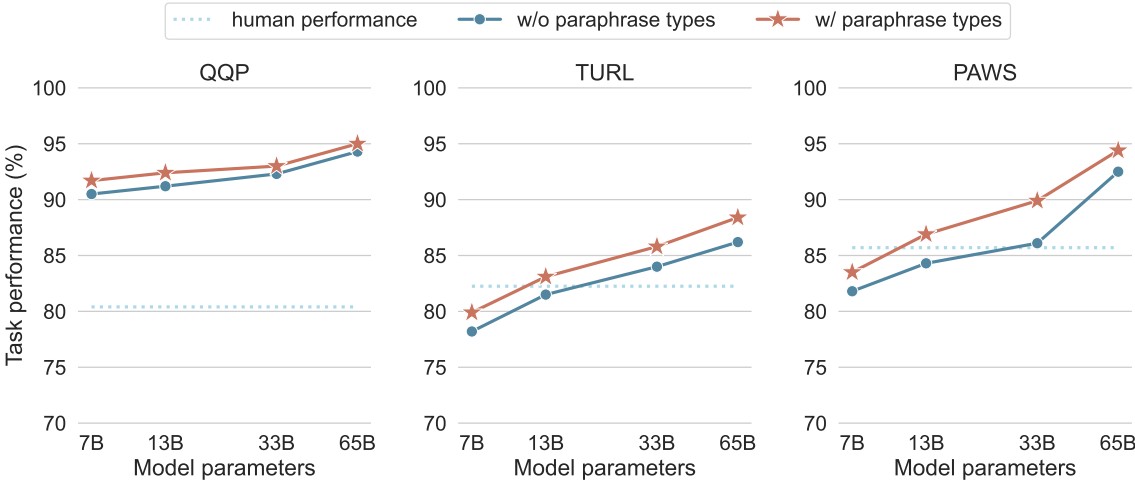

Figure 3: Task performance (accuracy) for different model sizes of LLaMA with and without learned paraphrase types against human performance as reported by the respective datasets or benchmarks.

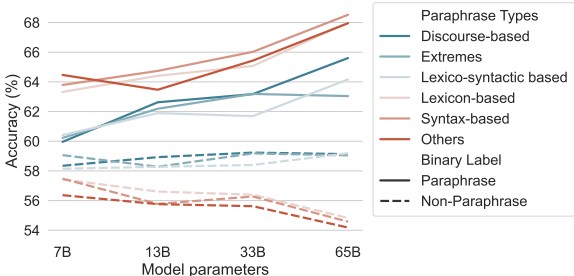

Figure 4: Accuracy for identifying paraphrase types of six high-level groups for different model sizes in the ETPC dataset using LLaMA.

## 4.2 Generation & Detection

**Q1.** *How much are paraphrase types already represented in language models?*

We test the ability of models to identify paraphrase types for different model sizes of LLaMA (in terms of model parameters). Therefore, we compose in-context prompts with few-shot examples and chain-of-thought (Wang et al., 2022; Wei et al., 2023). Prompt examples are shown in Figure 9 in Appendix A.6. We measure the accuracy of detecting the correct paraphrase type using the ETPC group categories. Figure 4 shows the results. While smaller models lead to overall low performance in identifying paraphrase types, scaling them increases the performance in identifying paraphrase types. As we scale the model, a divergence between paraphrases and non-paraphrases becomes more prominent for different types. We colored the three most divergent groups in red, i.e., lexicon-based changes, syntax-based, and others (mainly addi-

tions and deletions). One possible explanation is that phrases can be syntactically different and mean the same or have opposing meanings, leading to divergences. Larger models tend to learn the difference between paraphrases and non-paraphrases for individual types better. The overall performance, though, is relatively low, reflecting that LLMs also have difficulties classifying paraphrase types in general, meaning that they are unable to identify the particular changes that led to detection, even though they show good performance for the known *Binary* classification case (not shown here[4]).

**Q2.** *How well can language models perform type generation and detection when instructed to?*

**Generation.** We use BART and PEGASUS to perform Paraphrase Type Generation by adapting their last layer for token prediction. We assign each sample token-level label for generating substitutions of a paraphrase type.

$x$ Amrozi accused his brother, whom he called 'the witness', of deliberately distorting his evidence.

$\tilde{x}$ Referring to him as only 'the witness', Amrozi accused his brother of deliberately distorting his evidence.

$L(x)$ (26, 26, 26, 26, 0, 5, 0, 6, 25, 25, 25, 25, 25, 25, 25, 25, 25, 25, 25)

$L(\tilde{x})$ (6, 5, 5, 0, 25, 0, 0, 0, 0, 0, 26, 26, 26, 26, 0, 0, 0, 0, 0, 0)

---

[4]see https://paperswithcode.com/task/qqp

| Paraphrase Type | # Examples |
|---|---|
| **Morphology-based changes** | **975** |
| Derivational Changes | 186 |
| Inflectional Changes | 606 |
| Modal Verb Changes | 183 |
| **Lexicon-based changes** | **6 366** |
| Spelling changes | 628 |
| Change of format | 236 |
| Same Polarity Substitution (contextual) | 4 138 |
| Same Polarity Substitution (habitual) | 831 |
| Same Polarity Substitution (named ent.) | 533 |
| **Lexico-syntactic based changes** | **950** |
| Converse substitution | 43 |
| Opposite polarity substitution (contextual) | 15 |
| Opposite polarity substitution (habitual) | 4 |
| Synthetic/analytic substitution | 888 |
| **Syntax-based changes** | **731** |
| Coordination changes | 47 |
| Diathesis alternation | 161 |
| Ellipsis | 65 |
| Negation switching | 20 |
| Subordination and nesting changes | 468 |
| **Discourse-based changes** | **617** |
| Direct/indirect style alternations | 19 |
| Punctuation changes | 293 |
| Syntax/discourse structure changes | 305 |
| **Extremes** | **2 287** |
| Entailment | 81 |
| Identity | 1 782 |
| Non-paraphrase | 424 |
| **Others** | **5 742** |
| Addition/Deletion | 4 733 |
| Change of order | 857 |
| Semantic-based | 152 |
| Total | 16 813 |

Table 1: An overview of considered paraphrase types and their occurrences in the ETPC dataset.

Each label is mapped to a paraphrase type, and the tuple index corresponds to the tokenized word index. We balance the number of paraphrase types between training and validation sets, although this sometimes leads to low amounts of evaluation examples (e.g., $\frac{1}{4}$ examples for opposite polarity substitution). Thus, we consider only groups with at least 100 examples per type in their evaluation set.

We also test paraphrase type generation with ChatGPT-3.5 by formulating generation prompts as instructions (see Figure 9 for examples).

Table 2 shows the results. Both BART and PEGASUS show strong performance for generating paraphrase types in ETPC. BART outperforms PEGASUS in all metrics, particularly in ROUGE-L, suggesting that BART may be better suited for the task when contexts are longer. Although we expected fine-tuned ChatGPT to be superior over smaller models, it achieves higher BLEU scores but

lower ROUGE scores than BART. ChatGPT generates text that matches the reference at the n-gram level more precisely but might be missing out on covering other parts of the reference. This means the generated text might be very similar to some portions of the reference but does not capture the entirety or breadth of the reference content. BART and PEGASUS capture most of the content from the reference text, but how they present it (wordings, order) might differ from the reference. This means the generated text has a good recall of the critical content but may not have the exact phrasing or structure as the reference. Table 5 in Appendix A.4 shows additional in-context predictions of ChatGPT, showing that the default model is not able to generate paraphrase types well without fine-tuning. Although ChatGPT-3.5 reached the highest BLEU scores, smaller models have an edge when fine-tuned. All tested models can learn and generate paraphrase types to some extent. Still, there is much potential for improvement using more sophisticated methods to generate paraphrase types.

**Detection.** We test four autoencoder models on Paraphrase Type Detection by adapting their token-level representation with a linear layer to classify one of the 26 paraphrase types. We also fine-tuned ChatGPT-3.5 on the same task with prompt instructions. To estimate the accuracy of classifying higher-level perturbations, we group the 26 types into one of six groups (Table 1). Both the type and group scores are averaged over all occurrences in the sequence. For the entire sequence, we classify the binary label (i.e., paraphrase or not) using the aggregate representation of the model (e.g., `[CLS]`-token for BERT).

Table 3 shows the accuracy for paraphrase type detection on ETPC and paraphrase detection on QQP. DeBERTa achieves the highest performance of all encoder models across all categories for detecting paraphrase types, with 83.0 in the traditional binary case, 65.0 when detecting paraphrase types, and 67.9 when detecting the correct group. ChatGPT significantly outperforms the results of DeBERTa and when detecting paraphrase types, it achieves 11.8 percentage points higher results than DeBERTa. ERNIE 2.0 closely follows DeBERTa in binary detection with a score of 82.7 but trailed in type and group detection. Overall, the scores for paraphrase type and group are relatively low compared to the binary case, underlining the challenge for autoencoder models to grasp which lex-

| Model | BLEU | ROUGE-1 | ROUGE-2 | ROUGLE-L |
|---|---|---|---|---|
| BART | 46.3 | **56.2** | 34.9 | **54.2** |
| PEGASUS | 45.3 | 54.9 | 33.8 | 50.1 |
| ChatGPT-3.5 | **55.9** | 51.8 | 32.9 | 48.9 |

Table 2: Generation results of fine-tuned models on the ETPC dataset.

| Model | ETPC | | | QQP | |
| | Binary | Type | Group | w/o Types[*] | w/ Types |
|---|---|---|---|---|---|
| BERT | 74.1 | 58.7 | 60.0 | 89.3 / 72.1 | 91.6 / 88.6 |
| RoBERTa | 68.3 | 62.5 | 62.9 | 90.2 / 74.3 | 91.5 / 88.6 |
| ERNIE 2.0 | 82.7 | 64.2 | 65.9 | **90.9** / 75.2 | 92.4 / 89.7 |
| DeBERTa | 83.0 | 65.0 | 67.9 | 90.8 / **76.2** | **93.0 / 90.6** |
| ChatGPT-3.5 | **90.4** | **76.8** | **78.1** | 90.7 / 75.4 | 92.5 / 90.0 |

Table 3: Detection results for the ETPC (accuracy) and QQP (accuracy/F1) datasets. Models trained on ETPC are applied in QQP (w/ Types). [*]Official results for autoencoders (w/o Types) from GLUE leaderboard `https://gluebenchmark.com/leaderboard` for comparison. Best results in **bold**.

ical perturbation occurred. In the binary scenario, all models can distinguish between paraphrased and original content at a general level with good performance. However, this success diminishes in the presence of paraphrase types, corroborating that these models do not understand the intrinsic variables that have been manipulated yet.

**Q3.** *How does learning paraphrase types improve task performance of traditional paraphrase tasks with model scale*?

We test fine-tuned LLaMA in three binary paraphrase tasks, i.e., QQP, TURL, and PAWS, with two different settings: without paraphrase type instructions and with instructions using the ETPC dataset. For more details on the prompts used, see Figure 9. We also report the human performance of the respective dataset papers or benchmarks. The results reveal a positive trend when scaling LLaMA from 7B to 65B parameters. While scaling the model improves its baseline (w/o paraphrase types), the incorporation of paraphrase types leads to an increase in performances, achieving better-than-human results for all three datasets (Figure 3). Across datasets, the variation is lowest for QQP, which is also more than ten times larger (795k) than TURL (52k) or PAWS (65k). This analysis complements earlier findings that larger models are also more capable of paraphrase type tasks.

**Q4.** *What impact has learning paraphrase types on generating paraphrases*?

We test models previously trained on generating ETPC paraphrase types to generate paraphrases for the QQP dataset. Table 4 shows the results. Integrating paraphrase types into BART and PEGASUS leads to considerable performance improvements across all assessed metrics. When using ChatGPT with in-context instructions, considerable performance gains can be observed too but overall the results are lower than BART and PEGASUS, again underlining that specific tasks can benefit from smaller expert models. BART experiences a marked increase in ROUGE-L score from 41.8 to 44.2 and its ROUGE-1 score from 43.1 to 45.5, demonstrating improved results in paraphrase generation. Similar but overall less consistent improvements are also observed in PEGASUS, with BLEU increasing by 2.6 points and ROUGE-L score rise of 2.4 points. These results show that including paraphrase types positively affects performance in the QQP dataset. ChatGPT shows good performance in generating paraphrase types too, without fine-tuning, with increases of up to 3.2 percentage points (ROUGE-1). Both models drop in performance from ROUGE-1 to ROUGE-2 and again increase from ROUGE-2 to ROUGE-L, a finding consistent with related works (Li et al.,

| Model | BLEU | ROUGE-1 | ROUGE-2 | ROUGE-L |
|---|---|---|---|---|
| BART | | | | |
| + w/o paraphrase types | 44.7 | 43.1 | 25.3 | 41.8 |
| + w/ paraphrase types | **46.8** | **45.5** | **27.0** | **44.2** |
| PEGASUS | | | | |
| + w/o paraphrase types | 42.3 | 41.9 | 25.1 | 39.6 |
| + w/ paraphrase types | 44.9 | 43.6 | 26.8 | 42.0 |
| ChatGPT-3.5 | | | | |
| + w/o paraphrase types | 34.6 | 31.8 | 14.5 | 29.2 |
| + w/ paraphrase types | 34.7 | 35.0 | 16.9 | 37.4 |

Table 4: Generation results on the QQP dataset for trained models in the ETPC without and with prior paraphrase type generation training. Best results in **bold**.

2019; Liu et al., 2020b; Miao et al., 2019; See et al., 2017). Including paraphrase types overall leads to higher ROUGE-L scores, indicating higher recall and tendencies to perform better with longer contexts. Further investigations are necessary to conclude whether specific paraphrase types or additional training contributed to performance gains.

**Q5.** *What impact has learning paraphrase types on identifying paraphrases?*

We also test the transfer of models trained on detecting ETPC paraphrase types to the QQP task (right side of Table 3). For the QQP column, we fine-tuned models on QQP that have been previously fine-tuned with paraphrase types on ETPC[5]. Models trained on paraphrase types consistently outperform their counterparts in the binary setup (w/o Types) for the QQP dataset. For example, the accuracy of DeBERTa improved from 90.8/76.2 to 93.0/90.6 with the integration of paraphrase types. Similarly, when paraphrase types are incorporated, BERT's performance improves from 89.3/72.1 to 91.6/88.60. ChatGPT achieves comparable performance to autoencoders.

These results underscore the value of integrating paraphrase types in enhancing models' detection capabilities across different datasets and detection metrics. DeBERTa achieves the best performance with significant improvements when the model is trained to recognize paraphrase types. Although ETPC has a relatively small amount of examples, the performance benefits are clear and can potentially accelerate the development of new paraphrase detection methods using LLMs in the future.

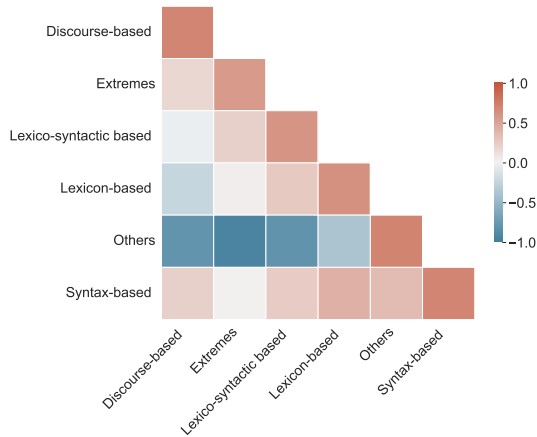

Figure 5: Rescaled Spearman correlations between paraphrase types in six higher-level families using word position deviation and lexical deviation. Correlations are normalized around the origin using mean $\mu$ and standard deviation $\sigma$ such that $\mu = 0, \sigma = 1$.

### 4.3 Type Correlations and Similarity

To show how different paraphrase types correlate, we compute word position deviation and lexical deviation (Liu and Soh, 2022b) of examples in the ETPC dataset. Next, we compute the Spearman correlations between these scores for all examples per paraphrase type to another and average over all examples per paraphrase type. Correlations are high overall, with an average of 0.89. We rescale the correlations in Figure 5 with $\mu = 0$ and $\sigma = 1$ to visualize the differences better. We include the correlation of all 26 paraphrase types in Figure 8 in Appendix A.2.

Within groups, lexicon-based changes have the highest correlation, followed by the "Others" category that contains "Addition/Deletion", "Change of

---

[5]We do not fine-tune ChatGPT on the full QQP dataset to reduce training cost. We sample 20% of training examples.

order", and "Semantic-based" changes. Between groups, we observe a higher-than-average correlation between syntax-based and lexico-syntactic-based changes, both containing syntactic components. The most prominent cases regarding group types are between "synthetic/analytic substitution" and "diathesis alternation"; and "opposite polarity substitution (contextual)" and "ellipsis" (see Appendix A.2 for more details). In summary, different paraphrase types show overall high correlations even between groups while lexicon-based and other changes correlate less.

### 4.4 Demo for Paraphrase Types

We provide an interactive chat-like demo on Huggingface[6] to generate paraphrase types interactively or automatically using the Python Gradio Client API. Figure 6 in the Appendix provides a screenshot of that tool.

## 5 Final Considerations

**Conclusion.** In this paper, we proposed Paraphrase Type Generation and Paraphrase Type Detection, two specific paraphrase type-based tasks. These tasks extend traditional binary paraphrase generation and detection tasks with a more realistic and granular objective. In the generation task, paraphrases have to be produced according to specific lexical variations, while in the detection task, these lexical perturbations need to be identified and classified. Our results suggest that the proposed paradigm poses a more challenging scenario to current models, but learning paraphrase types is beneficial in the generation and detection tasks. Additionally, both of our proposed tasks are compatible with existing ones. All models trained on paraphrase types consistently improve their performance for generation and detection tasks, with and without paraphrases. The shift from general paraphrasing to including specific types encourages the development of solutions that understand the linguistic aspects they manipulate. Systems that incorporate specific paraphrase types, can therefore support more interpretability, accuracy, and transparency of current model results.

**Future Work.** As the vast majority of datasets in paraphrasing do not account for paraphrase types, the first natural step is to evaluate how to include types in their composition. The expansion of cur-

rent datasets could take place either by automated systems (for large datasets) or by human annotators. In addition to generating new paraphrase-typed datasets, a prospective direction is to use large language models to paraphrase original content and qualitatively identify which paraphrase types these models learn during their training through humans. Although metrics such as BLUE and ROUGE have known deficiencies (e.g., high scores for low-quality generations), they are currently the standard practice in generation tasks. Thus, a metric incorporating paraphrase types with their location and segment length could provide a more accurate assessment of our proposed tasks.

## Limitations

As the use of paraphrase types in generation and detection tasks is still incipient, much work is required to establish this as the new paradigm in the field. To the best of our knowledge, our paper is one of the first contributions to define tasks for paraphrase types for automatic investigations, probe state-of-the-art models under these conditions, and verify the compatibility of specifically trained models with proposed and existing tasks. However, several points remain open to be explored. In this section, we go over some of them.

Two factors limit the experimental setup of our tasks: the datasets used and the considered metrics. Our analysis is based on ETPC to probe paraphrase types, so we are bounded to the limited number of examples of that dataset. In addition, we only test the transfer from paraphrase types to more general paraphrase tasks between ETPC and QQP. Thus, more diverse datasets must be proposed and explored so prospective solutions can be thoroughly evaluated. On the evaluation side, we still rely on standard metrics such as BLEU and ROUGE, which are known for their limitations (e.g., poor correlation with human preferences) and cannot account for paraphrase types, locations, or segment length in their score. A metric incorporating paraphrase types with their location and segment length would greatly support our experiments and results.

Even though we probe state-of-the-art models in our proposed tasks, no human study was conducted to establish a human baseline for comparison on paraphrase types. Particularly automated generation metrics such as ROUGE and BLEU work well for particularly paraphrase types, such as, syntax-changes but obviously have problems

---

[6]https://huggingface.co/spaces/jpwahle/paraphrase-type-generation

for lexicon-changes and lexico-syntactic changes. In future work, we are already exploring human annotation and alternative metrics to overcome issues resulting from word overlap. Part of the results obtained in Section 4 are limited to automatic solutions, and only traditional tasks involve a human baseline. Therefore, it is uncertain how much improvement current models still need to be comparable to human performance in generating and detecting paraphrase types.

## Ethics & Broader Impact

**Ethical considerations.** We understand that techniques devised for paraphrasing have many applications, and some of them are potentially unethical. As we push forward the necessity of paraphrase types, these can also be applied to generate more complex, human-like, and hard-to-detect paraphrased texts, which can be used for plagiarism. Plagiarism is a severe act of misconduct in which one's idea, language, or work is used without proper reference (Foltýnek et al., 2019; Kumar and Tripathi, 2013).

Large language models capable of mimicking human text are a reality (e.g., ChatGPT), and most of us still do not fully understand their reach. As already foreshadowed (Wahle et al., 2022a,b, 2021), paraphrasing using language models can lead to more undetected plagiarism, undermining the quality and veracity in several areas (e.g., academia, basic education, industry). Even though paraphrase types might encourage the development of even more sophisticated techniques that can potentially be incorporated into these models, we should not remain neutral. Therefore, artifacts to probe and understand what composes paraphrases in neural language models should be welcome instead of feared.

**Broader Impact.** The presented tasks in this work and its future solutions have the potential to benefit other areas aside from paraphrase generation and detection. In the following, we list a few applications that can be explored.

*Machine translation.* Paraphrase Type Detection can help identify paraphrase types in a translated text to identify areas where the translation is faithful to the original text. This task can also assist in identifying deficiencies in specific linguistic aspects between models and languages.

*Emotion analysis.* Paraphrase Type Generation could be used to express different emotions through multiple linguistic aspects. For example, the research could focus on comparing multiple versions of the same emotions and then estimate whether different linguistic concepts, such as negation, convey more or less emotion.

*Text summarization.* On top of Paraphrase Type Detection, researchers can build tools to identify where the summary preserves the original text's meaning, which parts of the text change, and how it impacts semantic preservation and coherence.

*Text generation.* Paraphrase Type Generation can support generating or paraphrasing stories using different paraphrase types to estimate which types lead to desirable attributes such as originality, tension, and character development.

## Acknowledgements

This work was supported by the DAAD (German Academic Exchange Service) - grant 9187215, the Deutsche Forschungsgemeinschaft (DFG, German Research Foundation) – grant 437179652, as well as the Lower Saxony Ministry of Science and Culture and the VW Foundation. Many thanks to Andreas Stephan for thoughtful discussions.

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

# A Appendix

## A.1 Frequently Asked Questions

### How do your tasks differ from syntactic and semantic learning?

The presented tasks are specifically designed to incorporate information about various linguistic aspects, not limited to syntax and semantics. They overcome shortcomings as demonstrated in our experiments in which one aspect is learned well (e.g., semantics - the subjects and objects and actions are the same), but others suffer (e.g., opposite polarity - the meaning was reversed). Investigating only semantics and syntax can lead to underrepresented paraphrase types, which are particularly problematic when shifting to new domains.

### The annotations for this task require experts. Is there an unsupervised approach that I can use?

While paraphrase types are diverse and have many categories, the annotation task is similar to a named entity recognition annotation, in which the text span (segment location) and entity (paraphrase type) must be annotated. High amounts of labels are also not a new challenge. Large annotation providers (e.g., Prodigy[7], Scale[8]) provide tools to simplify this process (e.g., one annotator finds three categories at a time). Also, tasks with more complex problems and descriptions seem to be more beneficial for future research (Mohammad, 2016; Rozovskaya and Roth, 2010). We are currently working on a semi-supervised method for assisted paraphrase-type generation using contrastive learning and reinforcement learning from human feedback to facilitate this task. Still, for evaluation purposes, we require an annotated test set or human raters.

### How can I use the tasks?

Our implementation is available on GitHub[9] and a demo is available through Huggingface.

## A.2 Details on Correlations

Figure 8 provides the detailed correlation of Figure 5 for each paraphrase type. While the overall group correlations are also represented here, some

notable differences exist. For example, direct and indirect style alternations more-than-average with the change of order. While writing style seems to have many components, a significant one seems to be the ordering of sentences. Usually, the most important keywords of a sentence remain the same, but their ordering can vary - one of the reasons why word-count-based metrics such as BLEU and ROUGE are still used in many tasks.

Another high correlation exists for contextual opposite polarity substitution and change of order. Opposite polarity substitutions are cases in which the meaning of a term is opposed to the original (e.g., "Johnson quickly **accepted** the proposal." and "Johnson **rejected** the proposal without hesitation."). Changing polarity can often include changing the term's position if an exact opposite term does not exist. Therefore, an order change can often be explained by contextual opposite polarity. However, suppose the polarity change is habitual. In that case, there is often no need for word order changes as the same concept can be explained at the same text position (i.e., the meaning remains - "Leicester **failed** in both enterprises" and "He **did not succeed** in either case"). Changing polarity also correlates with ellipses which are typically shorter versions of the same phrase.

Much lower-than-average correlations appear for converse substitution with addition and deletion and a format change. Converse substitution means the change of action from subject to object (e.g., "Sam **bought** a new car from John." and "John **sold** his car to Sam."). As already illustrated by this simple example, a converse substitution often requires a format change and additions/deletions to ensure that the subjects/objects are connected to the action in both cases.

While many more correlations exist across the spectrum (e.g., modal verb changes to negation switching or subordination and nesting changes to syntax/discourse structure changes), their nature appears to be due to both often appearing in the same sentences. The correlation analysis of this study serves as a starting point for further investigations but is limited in that only 17 668 total paraphrase types occur across 5 801 sentences.

Figure 7 shows the rescaled BERTScore similarity scores for the example in Figure 1 with three paraphrase-type perturbations, i.e., diathesis alternation, polarity substitution, and lexicon change. Since the example has no word position

---

[7] https://prodi.gy/
[8] https://scale.com
[9] https://github.com/jpwahle/emnlp23-paraphrase-types

**Paraphrase Type Generator**

This demo uses a fine-tuned ChatGPT-3.5 model to generate paraphrases given specific paraphrase types.

**How to use:**

1. Select one or many type of paraphrase from the dropdown menu.

2. Enter a sentence in the text box.

3. Click the "Submit" button or hit enter.

4. The application will generate a paraphrase of the input sentence based on the selected type.

---

💬 Chatbot

These outlined a theory of the photoelectric effect, explained Brownian motion, introduced his special theory of relativity—a theory which addressed the inability of classical mechanics to account satisfactorily for the behavior of the electromagnetic field—and demonstrated that if the special theory is correct, mass and energy are equivalent to each other.

[System: Syntax/discourse structure changes]

Another of the papers introduced Einstein's special theory of relativity, which addressed the inability of classical mechanics to account for the behavior of the electromagnetic field, and demonstrated that, if the special theory is correct, mass and energy are equivalent to each other.

---

Dropdown

Syntax/discourse structure changes ✕ | ✕ ▾

Textbox

---

Submit

Clear

Figure 6: An interactive demo to generate paraphrase types.

deviations, we expect the column maxima to be diagonal-shaped like in the segment "about her plan to change the constitution". However, particularly for the paraphrase types in question, the similarities between "She" and "The president", "has given" and "gave", and "talk" and "speech" are lower between them and sometimes inferior to other similarities for the same terms. We also tested the same example with "he/him" and "they/them" pronouns instead of "she/her" to verify potential biases towards gender (i.e., presidents are mainly male in the training data), but the scores were comparable.

We hypothesize that similarities between segments with paraphrase types have lower similarity, on average, than their non-paraphrase type counterparts. This suggests that paraphrase types are semantically more challenging to identify.

## A.3 Datasets

**Quora Question Pairs (QQP)** is a collection of approximately 400k pairs of questions extracted from Quora[10], a platform for general question and answers. This dataset is annotated to identify whether one question is a rephrasing of another (Wang et al., 2017). QQP is one of the largest and most estab-

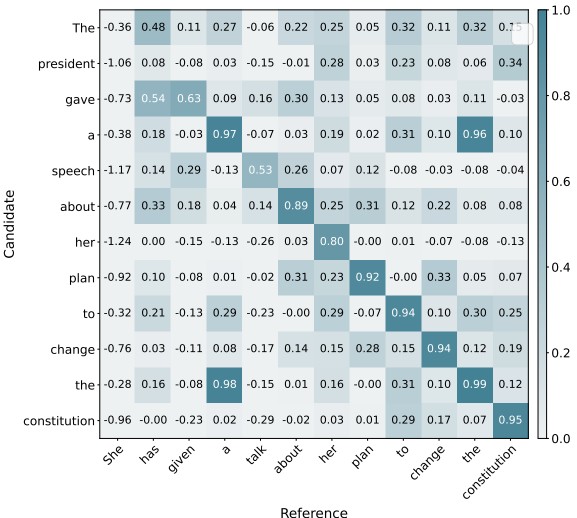

Figure 7: Rescaled BERTScore similarity for an example reference phrase and its paraphrased candidate.

lished paraphrase datasets in the community and therefore received particular attention throughout our experiments.

**Twitter News URL Corpus (TURL)** encompasses about 2.8 million pairs of human-authored sentences taken from Twitter[11] news (Lan et al., 2017).

---

[10] https://quora.com/

[11] https://twitter.com

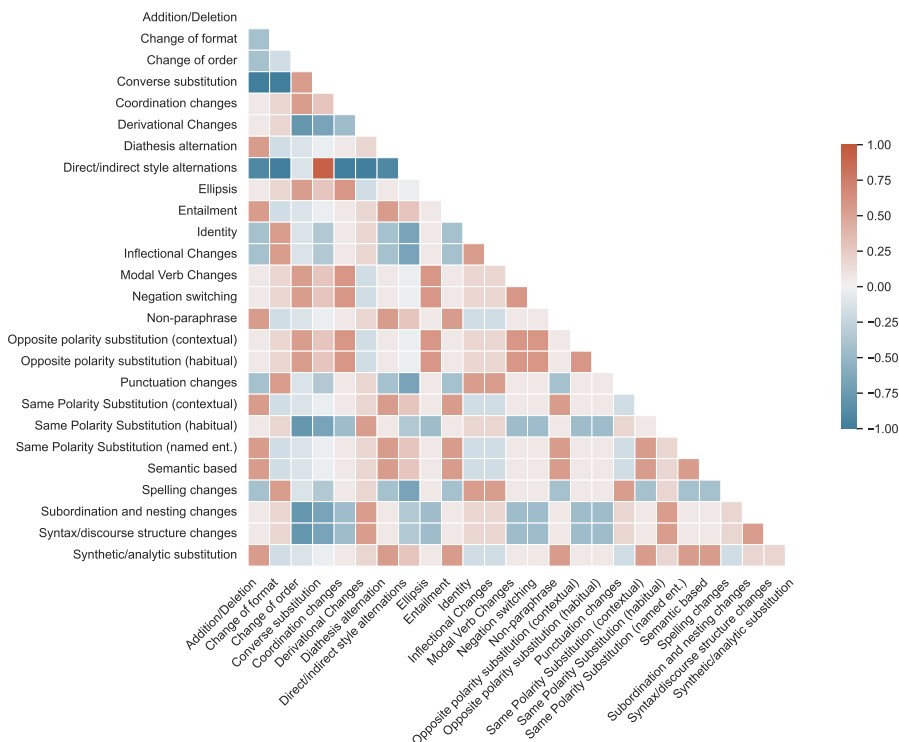

Figure 8: Rescaled Spearman correlations between paraphrase types using word position deviation, lexical deviation, BLEU, ROUGE, and BERTScore in the ETPC dataset. Correlations are normalized using mean $\mu$ and standard deviation $\sigma$ such that $\mu = 0, \sigma = 1$.

Annotations in the form of binary-type markings were provided by six individual raters. In our study, we recognized a paraphrase as positive based on a majority vote and neglected those pairs lacking majority consensus.

**Paraphrase Adversaries from Word Scrambling (PAWS)** comprises around 65k pairs of machine-generated texts (Zhang et al., 2019). These texts are obtained from Wikipedia and were created by implementing word reordering and back-translation strategies.

### A.4 Supplementary Results

Table 5 shows additional in-context prediction results for ChatGPT (i.e., prompts with few-shot examples). Both ROUGE and BLEU scores are much lower than those of fine-tuned models, showing that ChatGPT's default capability to generate paraphrase types is limited.

### A.5 Evaluation

Our detection experiments also included statistical significance tests of our results, namely results of Table 3. Following (Dror et al., 2018), we assess our results using a non-parametric sampling-free test, namely the Wilcoxon signed-rank test (Wilcoxon, 1992). The detection results between models are significant with p < 0.05.

### A.6 Prompt examples

Figure 9 shows some prompt examples used in our experiments. We rely on Wang et al. (2022) for few-show examples and prompt templates.

| | In-Context | | | Fine-Tuned | | |
|---|---|---|---|---|---|---|
| Model | BLEU | ROUGE-1 | ROUGE-L | BLEU | ROUGE-1 | ROUGE-L |
| BART | - | - | - | 46.3 | **56.2** | **54.2** |
| PEGASUS | - | - | - | 45.3 | 54.9 | 50.1 |
| ChatGPT-3.5 | 27.1 | 24.0 | 23.3 | **55.9** | 51.8 | 48.9 |

Table 5: Generation results on the ETPC dataset for **B**LEU, **R**OUGE-**1** and **R**OUGE-**L**.

**Paraphrase Type Generation**

Prompt: Given the following sentence, generate a paraphrase with the following types.

Sentence: And although the preconditions for recovery remain in place, it said the prospects for British exports were weaker than previously expected.
Paraphrase Types: Addition/Deletion, Inflectional ..

A: Although the preconditions for recovery remain in place, the prospect for external demand for UK output is weaker than previously expected

**Paraphrase Type Detection**

Prompt: Given the following two sentences, which of the paraphrase types are changed between them?

Sentence 1: I was wondering who they **might** be.

Sentence 2: It was unclear why that **could** be.

A: Modal Verb Change

Figure 9: Example prompts used for experiments on paraphrase type generation and detection.