# OpenReview forum: "Paraphrase Types for Generation and Detection"
_EMNLP/2023/Conference — EMNLP 2023 Main_

### Official Review · Reviewer_qoRy · 2023-08-05

**Soundness:** 3
**Typos Grammar Style And Presentation Improvements:** 1)	It is difficult to read figure 3 w…

**Excitement:**

2: Mediocre: This paper makes marginal contributions (vs non-contemporaneous work), so I would rather not see it in the conference.

**Paper Topic And Main Contributions:**

The work is focused on tasks of paraphrase type generation and paraphrase type detection where authors evaluate several PLM, and LLaMa utilizing Extended Paraphrase Typology Corpus (ETPC).  Authors show correlations between different paraphrase type groups and show how paraphrase detection(and generation) could help in the end task of paraphrase detection (binary). Work shows several examples and discusses paraphrase types from ETPC dataset highlighting the potential benefits on paraphrase identification (and other relevant tasks).

**Questions For The Authors:**

1)	Were any significance test done on the results and improvements?
2)	Why BART and PEGASUS (in table 2 and table 4) and not more recent decoder models like OPT?

**Reasons To Accept:**

1)	The paper is well motivated and it shows the need of paraphrase type generation and detection. The work could be beneficial for future works to consider paraphrase types while solving paraphrase identification task.
2)	A good set of PLM were evaluated on multiple paraphrase identification datasets
3)	The experiments are well motivated with corresponding research questions.

**Reasons To Reject:**

1)	The work intuitively makes that paraphrase type generation and detection would improve paraphrase identification task performance.
It could strengthen the impact of work if paraphrase generation (from different types) are shown to improve query rewriting/simplification and more downstream tasks.
2)	The proposed work seems more of an analyses of various PLM and LM on paraphrase type generation and detection. The correlation between models on different paraphrase types and groups is interesting but would strengthen the work if these can be shown with more concrete applications or their usefulness.
3)	LLaMa with paraphrase types show some improvement (a little unclear from the plots in figure 3, and if the difference is significant or not) in few shot prompt setting. It would be good to know if the difference becomes negligible or more if LLMs like LLaMa are instruction finetuned with lets say 10 examples of each paraphrase type.
4)	Some of the previous works have explored similar directions of Paraphrase Identification with types (see Paraphrase Identification with Deep Learning: A Review of Datasets and Methods). It would be good to discuss some of the previous works in this direction. The correlation analyses is given noticeable focus in the paper but could be very helpful to understand their applications as well.

**Reproducibility:**

4: Could mostly reproduce the results, but there may be some variation because of sample variance or minor variations in their interpretation of the protocol or method.

**Reviewer Confidence:**

3: Pretty sure, but there's a chance I missed something. Although I have a good feel for this area in general, I did not carefully check the paper's details, e.g., the math, experimental design, or novelty.

---

> ### Author Rebuttal · Authors · 2023-08-28
>
> We sincerely thank the reviewer for highlighting that this work is “well motivated and [...] shows the need of paraphrase type generation and detection” and that its findings “could be beneficial for future works”. We are also keen to hear that “the experiments are well motivated” and executed with multiple pre-trained language models and paraphrase identification datasets.
>
> Question 1: Were any significance test done on the results and improvements?
>
> Answer 1: Yes. We have conducted statistical tests for the detection task (see Appendix D.1). We will add them more prominently in the main body of the paper for the camera-ready version and consider performing statistical tests for the generation task too.
>
> Question 2: Why BART and PEGASUS (in table 2 and table 4) and not more recent decoder models like OPT?
>
> Answer 2: We chose BART and PEGASUS as they are usually a strong backbone of recent text generation and summarization systems (including in our previous experiments in correlated tasks). Additionally, as we propose new tasks, we wanted to establish a solid baseline for future contributions. This study’s goal is not to show/compare state-of-the-art performance but to confirm the hypothesis that paraphrase types improve detection and generation tasks.
>
> Thank you for mentioning the recent work “Paraphrase Identification with Deep Learning: A Review of Datasets and Methods” by Zhou et al. We will discuss and compare that work for the camera-ready version.
>
> Once again, thank you for your time and for engaging with the material in this paper and our response. We see the time

---

### Official Review · Reviewer_ow9w · 2023-08-05

**Soundness:** 3

**Excitement:**

4: Strong: This paper deepens the understanding of some phenomenon or lowers the barriers to an existing research direction.

**Paper Topic And Main Contributions:**

The work introduces two new paraphrase based tasks namely the paraphrase type detection and paraphrase type generation. The authors argue that existing semantic similarity based measures are limiting in paraphrase detection tasks as they do not consider the fine-grained linguistic variations in the paraphrases. The authors propose 7 different paraphrase types and study the effect of paraphrase types on paraphrase generation and paraphrase experiments. It is an interesting paradigm and the experimental results demonstrate positive impact of the proposed tasks and paraphrase types.

**Questions For The Authors:**

Did you consider existing augmentation literature like SSMBA, ROTOM and UDA which also introduce fine-grained linguistic variations as augmentation operators? I believe comparing with some of those methods might give more insights into the general category of operators needed for controlled paraphrasing.

Also please consider adding statistical significance tests for the various performance measures reported across tasks and datasets.

**Reasons To Accept:**

1. The work proposes new paradigm for paraphrase detection by introducing paraphrase types which cover diverse linguistic variations that could be introduced by paraphrasing a given piece of text. It leverages the fine-grained types annotated in ETPC corpus for the paraphrase type identification task. The proposed task is interesting and is of interest to the research community as paraphrase types help provide granular analysis of paraphrased text and also to control generated text.

2. The experiments carried out are extensive and not only demonstrate the performance gain on ETPC corpus but also demonstrate general applicability to duplicate detection datasets like Quora question pairs. The authors also evaluate the impact of incorporating paraphrase types for tasks of paraphrase identification and generation.

3. The authors experiment with diverse models of different scales, also providing insight into the emergent capabilities of large language models and also smaller models in transfer learning setting.

In summary an interesting preliminary work into new task of paraphrase type generation and detection which of interest to the community

**Reasons To Reject:**

1. I think statistical significance tests  along with reports of effect size are necessary to establish claims of superior performance.
2. Most paraphrasing datasets have short single sentence samples. More analysis on realistic cases of paraphrasing paragraphs where semantic context has to be maintained for longer sequences would help determine generality.
3. While it is appreciated that authors discuss limitations, it is not clear if the proposed paraphrase types are universally applicable to different datasets.  Though it is shown that including types helps in paraphrase detection on QQP the model trained on ETPC is applied to QQP without any type annotations for QQP. This may miss out some linguistic variations particular to QQP.

**Reproducibility:**

5: Could easily reproduce the results.

**Reviewer Confidence:**

5: Positive that my evaluation is correct. I read the paper very carefully and I am very familiar with related work.

---

> ### Author Rebuttal · Authors · 2023-08-28
>
> We would like to express our gratitude to the reviewer for their feedback and confident observations about our work. We want to underline the assessment that this work is “of interest to the research community as paraphrase types help provide granular analysis of paraphrased text and also to control generated text”. We are happy to hear that “the experiments carried out are extensive and demonstrate the performance gain on ETPC corpus and general applicability to duplicate detection datasets”.
>
> Re "Did you consider existing augmentation literature like SSMBA, ROTOM and UDA":
>
> Thank you for the interesting suggestions for data augmentation methods. We will investigate them thoroughly and add relevant comparisons to our related work and discussion.
>
> Re "Also please consider adding statistical significance tests":
>
> We have conducted statistical tests for the detection task (see Appendix D.1). We will add them more prominently in the main body of the paper for the camera-ready version.
>
> Once again, thank you for your time and for engaging with the material in this paper and our response. We see the time and thought you have put into this and very much appreciate it.

---

### Official Review · Reviewer_aya4 · 2023-08-07

**Soundness:** 3

**Excitement:**

3: Ambivalent: It has merits (e.g., it reports state-of-the-art results, the idea is nice), but there are key weaknesses (e.g., it describes incremental work), and it can significantly benefit from another round of revision. However, I won't object to accepting it if my co-reviewers champion it.

**Missing References:**

Please consider citing the following papers, which are closely related to your work.

1. PIP: Parse-Instructed Prefix for Syntactically Controlled Paraphrase Generation; Yixin Wan, Kuan-Hao Huang and Kai-Wei Chang; Findings of ACL 2023

2. Improving Large-scale Paraphrase Acquisition and Generation; Yao Dou, Chao Jiang, Wei Xu; EMNLP 2022

**Paper Topic And Main Contributions:**

This paper studies the task of paraphrase identification and generation. Unlike traditional binary settings, the authors consider 26 fine-grained paraphrase types.

The authors experiment with several baseline methods for both tasks and demonstrate that incorporating fine-grained paraphrase types can help boost paraphrase generation and detection tasks under binary labels.

**Questions For The Authors:**

Please see above.

**Reasons To Accept:**

1. The motivation for incorporating fine-grained types into paraphrase generation and identification makes sense, though I am unclear on some details about the task setup.

2. I think the experiment setup is reasonable and well-executed. The author conducts pretty comprehensive experiments to support their claim.

**Reasons To Reject:**

1. I have a hard time following the paper. Here are some of my confusions:

    - In line 193 and line 211, when you say give a phrase x, do you mean a sentence x or a multi-word phrase x?

Confusions about paraphrase type generation:

- The paraphrase type generation task is a sentence-level task or phrase-level task? From the example in Figure 10 in the appendix, it seems to be a sentence-level task. Then why in line 206, paraphrase *segments* are mentioned?
- Assuming the paraphrase type generation is a sentence-level task, given an input x and a set of paraphrase types, assume the system returns x$^*$. How do you make sure  x$^*$ has the designated paraphrase types? I am unsure if using BERTScore / BLEU / ROUGE is enough here. Maybe you can do a small portion of manual inspection to see if the output has the paraphrase types, if not using some automatic method?

Confusions about paraphrase type detection:

- From Figure 10 in the appendix, it seems the input to this task are two sentences.  Will the two "re-phrased word pairs" be provided? Or they are the system outputs, in addition to the system output? What if there are multiple re-phrased word pairs?

**Reproducibility:**

3: Could reproduce the results with some difficulty. The settings of parameters are underspecified or subjectively determined; the training/evaluation data are not widely available.

**Reviewer Confidence:**

3: Pretty sure, but there's a chance I missed something. Although I have a good feel for this area in general, I did not carefully check the paper's details, e.g., the math, experimental design, or novelty.

---

> ### Author Rebuttal · Authors · 2023-08-28
>
> We are happy to hear that the motivation for the proposed paraphrase-type tasks is clear and supported through “comprehensive experiments” and an “experiment setup [that] is well-executed”.  We thank the reviewer for their feedback and clarify questions in the following.
>
> Re: “The paraphrase type generation task is a sentence-level task or phrase-level task?”:
>
> Yes, paraphrase type generation and detection are sentence-level tasks. By “segments”, we mean subparts of the sentence which do not necessarily convey enclosed meaning. We use segments to identify the location of where a paraphrase happened. We will adjust the use of “sentence”, “phrase”, and “segment” to avoid any interpretation issues.
>
> Re: “I am unsure if using BERTScore / BLEU / ROUGE is enough here.”:
>
> We agree that BERTScore, BLEU, and ROUGE are limited in capturing semantic similarity but are also commonly used for evaluating generative tasks. As the community often uses these metrics as a proxy in paraphrasing tasks, we decided to keep them so our findings/results can be easily compared in future research. Further, we inspected a few examples non-systematically throughout the experiments to confirm the expected model behavior.
>
> Re: “From Figure 10 in the appendix, it seems the input to this task is two sentences. Will the two "re-phrased word pairs" be provided?”:
>
> For detection, the system receives two sentences with a specific indication about the location of the paraphrase. Therefore, multiple paraphrases can occur within the same sentence but at different locations. The system returns the single type or group of the paraphrase (e.g., Inflection Change, Lexicon Change). We will clarify this in the figure.
>
> Thank you for the suggestions on additional related work. We will include these references in our related work for the camera-ready version of the paper. Once again, thank you for your time and for engaging with the material in this paper and our response. We see the time and thought you have put into this and very much appreciate it.

---

### Meta-Review · Area_Chair_xZfW · 2023-09-24

**Recommendation:** 4

**Metareview:**

This paper studies the task of paraphrase identification and generation. Unlike traditional binary settings, the authors consider 26 fine-grained paraphrase types (most of which proposed by Kovatchev, Marti, Salamo 2018) -- this is a promising direction of research on paraphrase, and addresses the issue that the definition of paraphrase is often not precise enough. Authors conducted some interesting analyses on four paraphrase datasets, with Llama and several other LLMs.

Author's responses also raised some confidence -- two of the reviewers are happy with the author responses. We hope the authors will make improvements based on the review comments for their camera-ready, if the paper gets accepted. We also want to ask the authors to discuss more thoroughly the related work and limitations.

---

### Decision · Program_Chairs · 2023-10-07

**Decision:**

Accept-Main

**Comment:**

This paper studies the task of paraphrase identification and generation. Unlike traditional binary settings, the authors consider 26 fine-grained paraphrase types (most of which proposed by Kovatchev, Marti, Salamo 2018) -- this is a promising direction of research on paraphrase, and addresses the issue that the definition of paraphrase is often not precise enough. Authors conducted some interesting analyses on four paraphrase datasets, with Llama and several other LLMs.

Author's responses also raised some confidence -- two of the reviewers are happy with the author responses. We hope the authors will make improvements based on the review comments for their camera-ready, if the paper gets accepted. We also want to ask the authors to discuss more thoroughly the related work and limitations.